# Tissue Engineered Esophageal Patch by Mesenchymal Stromal Cells: Optimization of Electrospun Patch Engineering

**DOI:** 10.3390/ijms21051764

**Published:** 2020-03-04

**Authors:** Silvia Pisani, Stefania Croce, Enrica Chiesa, Rossella Dorati, Elisa Lenta, Ida Genta, Giovanna Bruni, Simone Mauramati, Alberto Benazzo, Lorenzo Cobianchi, Patrizia Morbini, Laura Caliogna, Marco Benazzo, Maria Antonietta Avanzini, Bice Conti

**Affiliations:** 1Department of Drug Sciences, University of Pavia, 27100 Pavia, Italy; silvia.pisani01@universitadipavia.it (S.P.); enrica.chiesa01@gmail.com (E.C.); ida.genta@unipv.it (I.G.); bice.conti@unipv.it (B.C.); 2Department of Clinical, Surgical, Diagnostic & Pediatric Sciences, University of Pavia, IRCCS Policlinico S. Matteo, 27100 Pavia, Italy; stefania_croce186@yahoo.it (S.C.); l.cobiachi@smatteo.pv.it (L.C.); 3Department of Paediatric Oncoaematology, IRCCS Policlinico S. Matteo, 27100 Pavia, Italy; elisa.lenta@yahoo.it (E.L.); ma.avanzini@smatteo.pv.it (M.A.A.); 4Department of Chemistry, University of Pavia, 27100 Pavia, Italy; giovanna.bruni@unipv.it; 5Department of Surgery, Otolaryngologist section, IRCCS Policlinico S. Matteo, 27100 Pavia, Italy; s.mauramati@smatteo.pv.it (S.M.); m.benazzo@smatteo.pv.it (M.B.); 6Department of Surgery, Medical University of Vienna, 1090 Vienna, Austria; alberto.benazzo@meduniwien.ac.at; 7Department of Diagnostic Medicine, IRCCS Policlinico S. Matteo, 27100 Pavia, Italy; p.morbini@smatteo.pv.it; 8Orthopaedic and Traumatology, IRCCS Policlinico San Matteo, 27100 Pavia, Italy; l.caliogna@smatteo.pv.it

**Keywords:** electrospinning, patch engineering, temperature induced precipitation, porcine mesenchymal stem cells

## Abstract

Aim of work was to locate a simple, reproducible protocol for uniform seeding and optimal cellularization of biodegradable patch minimizing the risk of structural damages of patch and its contamination in long-term culture. Two seeding procedures are exploited, namely static seeding procedures on biodegradable and biocompatible patches incubated as free floating (floating conditions) or supported by CellCrown^TM^ insert (fixed conditions) and engineered by porcine bone marrow MSCs (p-MSCs). Scaffold prototypes having specific structural features with regard to pore size, pore orientation, porosity, and pore distribution were produced using two different techniques, such as temperature-induced precipitation method and electrospinning technology. The investigation on different prototypes allowed achieving several implementations in terms of cell distribution uniformity, seeding efficiency, and cellularization timing. The cell seeding protocol in stating conditions demonstrated to be the most suitable method, as these conditions successfully improved the cellularization of polymeric patches. Furthermore, the investigation provided interesting information on patches’ stability in physiological simulating experimental conditions. Considering the in vitro results, it can be stated that the in vitro protocol proposed for patches cellularization is suitable to achieve homogeneous and complete cellularizations of patch. Moreover, the protocol turned out to be simple, repeatable, and reproducible.

## 1. Introduction

There are several conditions, both congenital and acquired (malignant or benign tumor, in some forms of atresia, caustic lesion, and severe reflux), for which esophageal replacement is needed [1]. The surgical intervention remains the preferred treatment, however alternative options are essential for restoring esophageal continuity and functionality [2,3,4,5]. These alternative approaches may be associated with several complications, such as inflammatory response, scar deposition, stricture formation, and in the most serious cases, they could cause the morbidity of surrounding tissues [6].

In the last few years, tissue engineering (TE) has emerged as a promising solution for the replacement of physiological esophageal functions [7,8,9]. This field of science has deeply benefited from meaningful progresses in biomaterial science, nanotechnologies, bioreactor technology, molecular biology, and stem cells discovery [10,11].

Material of animal origin, such as decellularized matrices (DM), and synthetic and natural biomaterials were extensively studied for producing scaffolds useful to guide tissue regeneration. The scaffolds are turning into sophisticated structure for local delivering of active agents (such as proteins, genes, growth factors, antibiotics, and anti-inflammatory drugs) in their free form or loaded in micro- and nano-carriers [12].

Decellularized scaffolds from animal esophagus were recently studied, demonstrating that the decellularization protocol can be standardized and reproducible [7,13]. Luc and coll. successfully demonstrated that decellularized scaffolds from porcine esophagus were re-cellularized with porcine mesenchymal stem cell and promising results have been observed after in vivo implantation [12].

In this regard, the issue related to rejection or host immune response activation has been mitigated producing scaffold based on biocompatible and absorbable polymers. The research in this area is very dynamic, having a high interest in finding multidisciplinary approaches for solving transplant issues [8,14,15,16,17,18,19]

Several cutting-edge technologies have been exploited in the last years, such as electrospinning, 3D printing, and 3D bioprinting, for producing biodegradable and biocompatible scaffolds having structures that resemble the natural extracellular matrix (ECM). The electrospinning is a fabrication technology based on high electric fields that can be used to produce fibers ranging from the submicron to nanometers size. The most interesting aspect, supporting electrospun scaffold application in TE, is that their structure resembles the scale and three-dimensional arrangement of collagen fibrils in the ECM supporting cell adhesion, migration, proliferation, and making them crucial in several biological and medical applications [20,21].

Cells are another essential component in regenerative medicine. Mesenchymal stem cells (MSCs) are considered as primary adult stem cells with a high proliferation capacity, wide differentiation potential, and immunosuppression properties, which make them unique for regenerative medicine and cell therapy. The well-documented self-renewal and differentiation capacities make them a promising and valuable approach for cellularization of electrospun scaffolds in TE [22,23]. It has been recently reported that the MSCs may be a useful source of factors triggering cell proliferation and regeneration of damaged tissue [24]. Moreover, MSCs can be retrieved from different sources and in vitro expanded until an adequate number of cells are reached for in vivo application.

The present research project has been developed by a multidisciplinary approach (pharmaceutical technologist, chemists, surgeons, and biologists) with the aim of optimizing patch design, and its cellularization procedures. These last involved cells retrieve by biopsy from pig animal, porcine mesenchymal stem cells (p-MSCs) cells expansion and characterization, and final seeding. The seeding phase has been established to be crucial to achieve the complete cellularization of prototype in suitable timing for the following implantation in animal model.

The main goal of work was to locate a simple, reproducible protocol for uniform seeding and cellularization of biodegradable patch minimizing the risk of structural damages of patch and its contamination in long-term culture. Two seeding procedures are exploited, namely static seeding procedures on biodegradable and biocompatible patches incubated as free floating or supported by CellCrown^TM^ insert (fixed conditions) and engineered by porcine bone marrow MSCs (p-MSCs). Scaffold prototypes having specific structural features concerning pore size, pore orientation, porosity, and pore distribution were exploited using two different techniques, such as temperature-induced precipitation (TIP) and electrospinning (EL). The investigation on different prototypes allowed achieving several implementations in terms of cell distribution uniformity, seeding efficiency, cellularization timing, repeatability, and reproducibility.

## 2. Results

### 2.1. p-MSC Characterization

In culture, p-MSC showed the typical spindle shape morphology (Figure 1a). The flow-cytometry analyses of surface antigen expression showed that p-MSC were positive for CD90, CD29, and CD105; and negative for CD45 and CD11b, which are typical markers of hematopoietic cells (Figure 1b). In addition, p-MSC differentiated into osteoblasts as demonstrated by the histological detection of calcium depositions positive for Alzarin Red, and into adipocytes as shown by the morphological appearance of lipid droplets stained with Oil Red O (Figure 1c).

Percentages of antigen surface expression of pBM-MSCs after 30 days of culture have been plotted in Figure 2. No significant differences have been noted after 30 days and this incubation timing which was selected as useful for homogenous cellularization.

### 2.2. Patches Physicochemical Characterization

#### 2.2.1. Scanning Electron Microscopy (SEM)

The images obtained by SEM (Figure 3a,b) did not show any significant morphological differences between both EL-Ms layers. The fiber size was in the 700–800 nm range, maintaining random orientation. SEM images of cross section allowed highlighting an interconnected porous structure achieved by fiber entanglement. The thickness of the bilayer sample was 41–81 μm as measured by the cross section analysis (Figure 3c).

On the contrary, TIP-Fs layers showed significant morphologic differences: PLA:PCL 85:15 layer (Figure 3d) showed smoother and less porous surface, while the PLA-PCL 70:30 layer exhibited significant macro sized porosity throughout the whole patch surface random orientation (Figure 3e). SEM images of cross section (Figure 3f) displayed an interconnected porous structure, likely due to the solvent sublimation in the freeze-drying process. Patch thickness resulted to be ~660 μm.

#### 2.2.2. Contact Angle (θ) Measurement

TIP-Fs showed significant differences in contact angle values, PLA-PCL 85:15 layer showed highest θ values with both solutions tested at t_0_ (77.31° with DMEM and 73.75° with ASS). The values were 68.43° with DMEM-LG and 74.15° with ASS after contact for 1 min.

PLA-PCL 70:30 layer showed lower θ values at t_0_ (62° in DMEM-LG and 60.22° in ASS) and greater decrease after 1 min contact (43.6° in DMEM-LG and 52.3° in ASS). This evidence could be attributed to the distinct layer structure, the greater porosity of PLA:PCL 70:30 layer facilitates liquid absorption and its permeation in the patch.

Electrospun samples showed θ values lower than 10° in each tested fluid, both at time zero and at contact timing 1 min, demonstrating their optimal wettability. The results demonstrated the fabrication process and consequently their resulting structure significantly affected patches wettability, since TIP-Fs and EL-Ms prototypes are based on the same polymer combination.

#### 2.2.3. In Vitro Degradation Test

Results of in vitro degradation tests are collected in Figure 4a,b. Sterilization by γ-irradiation did not significantly affect polymer Mw, Mn, and PI, as shown by the comparing of t_0_ and t_0_/γ-irr data. pH of culture medium was monitored during the degradation test, pH 8.0 was measured at the beginning of test and it slightly boosted at 9.0 after 7 days, no further changes have been detected. A similar trend was observed for all samples, and the pH shift observed at day 7 was attributed to the amino acid L-glutamine content in DMEM-LG.

Considering GPC results, TIP-Fs prototypes showed a significant decrease in Mw, Mn, and PI starting from day 15 reaching 57% of Mw reduction after 45 days of incubation. EL-Ms degradation rates were significantly slower comparing with TIP-Fs (Figure 4b), 31% of M_w_ reduction was detected at day 46 reaching ~44% at day 60. PI values gradually decreased achieving ~1 at day 60. This behavior is consistent speculating a gradual release of small soluble oligomers arising from polymer chain scission.

Figure 4c and d shows the decrease of M_w_, M_n_, and PI values for TIP-Fs and EL-Ms prototypes incubated in ASS at 37 °C. The experiment was performed on samples previously incubated with p-MSCs, in 10% DMEM-LG for 30 days, which was established as time zero. pH values of ASS maintained at 8.03 ± 0.25 throughout the period considered.

TIP-Fs Mw value dropped to 50 kDa at day 30 (Figure 4c) and to 35 kDa at day 60, corresponding to 37.5% and further 30% of M_w_ reduction, respectively. The TIP-Fs Mw reduction of samples incubated in ASS was more gradual reaching 74% of reduction at day 60; moreover, no significant variations of f Mn values were observed over the time.

GPC results demonstrated that EL-Ms degradation is significant slower with respect to TIP-Fs in the same experimental conditions (Figure 4d). Indeed, EL-Ms M_w_ value decreased to 68 kDa after 60 days incubation, corresponding to 25.5% and 31% of M_w_ reduction with respect to M_w_ at time zero and to pristine M_w_ of t_0/y_-irradiated EL-Ms.

Degradation rate in ASS seemed to be slower and more gradual; this evidence could be justified by hypothesizing the solubilizing of the small oligomers in the aqueous medium.

To conclude, the degradation performances of TIP-Fs and EL-Ms were not affected by incubation with p-MSCs (Figure 4a–d).

### 2.3. Biological Characterization

#### 2.3.1. p-MSC Seeding on Floating Matrices

Data reported in Figure 5 showed poor cell proliferation at day 13 corresponding to 0.7% and 10.6% for TIP-Fs and EL-M prototypes, respectively. A significant increase, ranging between 27.9 and 59.9%, was detected at day 45 for EL-Ms and at 60 days for TIP-Fs. MTT was carried out on well bottoms at day 15 and 30, and showed greater cell viability percentage with respect to values of TIP-Fs and EL-Ms samples.

However, the cell viability values greatly boosted at day 45 for both the tested samples; the failure observed at day 60 for EL-Ms samples was justified speculating an insufficient available surface area for further cell proliferation.

The figures collected in Figure 6I(a,b) are consistent with MTT data, a low number of cells attached on TIP-Fs sample surfaces was observed at day 15 and 30, however samples were well populated, Figure 6I(c,d), at day 45 and 60. At each established timing, p-MSCs seeded on TIP-Fs were subjected to treatment with DAPI for further corroboration with MTT data, Figure 6I(e–h).

Figure 6II(a–c) and d shows SEM and confocal images of the EL-Ms cellularized with p-MSCs. It is possible to notice that both cells homogeneously proliferated on electrospun samples (Figure 6II(a,b)) after 15 and 30; moreover, cells appearing well stretched and connected one with the other. Following, the sample was well proliferated at day 45 (Figure 6II(c)) comparing to Figure 6II(d) (day 60). However, the cells’ high proliferation observed in the first 30 days gradually decreased because of their detachment making the patch surface visible. Confocal analyses are quite consistent, Figure 6II(e–h)

#### 2.3.2. Cell Seeding Using CellCrown^TM^ System

The using CellCrown^TM^ system substantially reduced patches floating during incubation; the high patch mobility observed in floating conditions was identified to be a crucial factor affecting cell proliferation and growth. The results reported in Figure 7a are related to patches incubated with p-MSCs at day 30, which was pinpointed as the timing required for reaching the complete patch cellularization. Cell viability values were >100% for both TIP-Fs and EL-Ms; however, cell viability values obtained for TIP-Fs at day 30 were significantly higher (360%) than TIP-Fs sample in floating conditions (47%). Moreover, the sample immobilization had a positive effect on cell attachment reducing their sliding on well bottom immediately after seeding (0.4 days).

Figure 7b(I) shows SEM images of TIP-Fs incubated, on immobilized samples, with p-MSCs at day 30. It is clear that with the uniform cell growth on sample surface, cells are able to attach, proliferate, and grow, creating connections and cell communication. Figure 7b(II) shows confocal microscope images obtained staining the cellularized TIP-Fs samples with DAPI dye. The images corroborate cell viability results, cells seeded on fixed samples were greatly cellularized and the number of p-MSCs was visibly greater than floating samples at day 30 (Figure 7b(II)). Consistently, Figure 7b(III) shows SEM image of the EL-Ms samples incubated in the same experimental conditions. Cellularization of EL-Ms was poorly uniform compared with TIP-Fs, indeed, the patch fibrous structure was clearly visible in same areas. The using of Cellcrow^TM^ constrained sample folding, a phenomenon that could promote cell detachment, or in the worst case, their death. The high cell growth was further proved by confocal analysis (Figure 7b(IV)).

The effect of seeding conditions on cell attachment was investigated performing an MTT test after 6 h (Figure 7c). Cell viability percentage of fixed TIP-Fs sample was 94.51%, a significantly greater value than that obtained for floating TIP-Fs sample (38.99%). The trend was comparable for fixed EL-Ms samples, which showed cell viability values of ~82% for the immobilized samples compared to 22.28% for the floating matrices. The 6-h adhesion test allowed highlighting how the cells seeding and their attachment are promoted by patch immobilization.

It is possible to conclude that using the CellCrown^TM^ system, the attachment of cells and their subsequent proliferation were significantly improved by minimizing the sliding of cell suspension and patches motions, which may result in detachment of cells and deposition on the well bottom. The differences in viability percentages between TIP-Fs and EL-Ms led to state that the cell attachment, and their proliferation and growth were successfully improved on electrospun prototype (TIP-Fs) compared with EL-Ms. The proliferation, migration, and differentiation of cells within the scaffold are mediated by the initial cell adhesion, which depends on pore size, pore size distribution, porosity percentage, and pore orientation. The preservation of a balance between the surface area for cell attachment and optimal pore size for cell migration is crucial in the complex cellularization process.

## 3. Discussion

Patch cellularization may be achieved by different seeding methods. Passive static seeding is the simplest cell culture method. It involves pipetting a cell suspension directly onto a patch surface following incubation of the seeded construct with media to allow cell attachment. The use of commercially available biological glues such as fibrin or fibronectin was investigated in order to maximize cell adherence efficiency [24,25,26]. As an alternative method, dynamic seeding systems were widely studied for increasing the cell seeding efficiency, other than cell distribution uniformity, and their penetration in scaffold architecture. The two main methods of dynamic seeding are rotational seeding exploiting hydrostatic forces [27,28,29] and vacuum seeding, which create pressure differentials able to force cell suspension diffusion through the patch [30]. Comparison between static and dynamic cell seeding in different culture systems sometimes led to controversial results. For example, Beskardes et al. compared dynamic and static cell seeding systems in bone tissue cultures, and they concluded that exclusively the static seeding method combined with perfusion culture could enhance cell viability and osteogenic differentiation [31,32].

In the present study, biodegradable and biocompatible polymeric patches addressed to esophagus regeneration were prepared with two different techniques (temperature-induced precipitation and electrospinning) in order to investigate how patch topography and architecture could influence cell adhesion and proliferation. Moreover, the chemical patch stability was tested in experimental conditions, simulating the in vivo operative ones. As long as patch material is concerned, several studies can be found in the literature proposing biomaterials such as poly(glycolic acid) meshes, polycarbonate, and polyurethane-based electrospun matrices, silicon meshes, polylactid acid, and polycaprolactone patches [24,33,34,35].

Bilayer patches made of polylactide-*co*-polycaprolactone (PLA- PCL) with diverse co-polymer ratios and concentrations were prepared by the temperature-induced precipitation method and by electrospinning. The patch preparation protocols have been discussed in previous works, together with their preliminary physicochemical characterization [20,35]. In the present work, the patch physicochemical characterization was deepened, and more focused on patch functional properties in the esophagus regeneration.

The results demonstrated that both TIP-Fs and EL-Ms structures present specific porosity, pore size, pore distribution, and orientation. The rationale was to orient the first layer towards esophagus lumen, while the more porous layer would be the external surface promoting cell growth and differentiation into muscular mucosa. Patch surface hydrophilicity plays an important role in the interaction with biologic fluids, and their topography depends on method exploited for producing the prototype; considering contact angles values, the wettability of TIP-Fs resulted poor comparing with the EL-Ms samples.

The polymer used for patch preparation is biodegradable and biocompatible; they are designed to be reabsorbed after in vivo implantation in a fixed time span that should be synchronized with tissue regrowth.

The choice of p-MSC for patch cellularization was supported by the well-known renewal and differentiation properties of these cells. In these terms, p-MSCs were here proposed as a means to favor tissue reconstruction. Our results demonstrated that p-MSCs were able to adhere, proliferate, and colonize both type of patches, confirming their biocompatibility. The results obtained by cell seeding on floating TIP-Fs and EL-Ms were significantly different in the first 30 days of incubation; in particular, cell adhesion and proliferation on TIP-Fs was very poor. The limited adhesion and proliferation were attributed to the floating of patch in the medium, its rolling during the incubation, and the turbulent conditions caused during the medium exchanges. Cell seeding in static conditions allowed achieving greater cell viability percentage in shorter times (30 days incubation), both on TIP-Fs and EL-Ms. The results demonstrated that cell growth with CellCrown^TM^ system was significantly superior. Moreover, TIP-Fs resulted to be more suitable for cell adhesion and proliferation than EL-Ms.

TIP-Fs and EL-Ms resulted in different degradation profiles, probably for the structure that characterized each type of patch. However, the incubation with p-MSC did not significantly modify their degradation behavior. The further incubation in ASS did not affect patch stability, the results demonstrated that the prototypes were enough stable in physiological simulated conditions for 2 months. Of course, these in vitro results can only give a stability indication that must be evaluated in vivo on porcine animal models.

## 4. Materials and Methods

### 4.1. Materials

Purasorb PLC 85:15 (L-lactide/caprolactone copolymer, molar ratio 85:15) Mw 180 kDa was from Purac Biomaterials, Netherlands; RESOMER LC 703 S (L-lactide/caprolactone copolymer, molar ratio 70:30) Mw 160 kDa was from EVONIK Nutrition and Care GmbH, 64275 Darmstadt, Germany. In the manuscript the copolymers will be identified as PLA-PCL (L-lactide/caprolactone copolymer) at ratio 85:15 and 70:30.

Tetrahydrofuran (THF) analytical grade 99.5% supplemented with butylated hydroxytoluene (BHT, 100–300 ppm), *N*,*N*-dimethylformamide (DMF, analytical grade 99.8%), dichloromethane (DCM, analytical grade 99.9%), 1,4-dioxane (analytical grade > 99.8%) were from Carlo Erba, Milano, Italy; dimethyl sulfoxide (DMSO, 99.94%) was from Sigma Aldrich, St. Louis, USA. Cell separation medium, Ficoll 1.077 g/mL (Lympho prep, Nycomed Pharma, Oslo, Norway). Uncoated polystyrene culture flasks (Corning Costar, Celbio, Milan, Italy). Complete p-MSC culture medium: Dulbecco’s Modified Eagle Medium low glucose (DMEM-LG) supplemented with GlutaMAX^TM^ (Gibco, Paisley, UK), gentamicin (Gibco) and Mesencult (Mesenchymal Stem Cell Stimulatory Supplements, StemCell Technologies, and Vancouver, Canada). Trypsin-EDTA was from Euroclone (Milan, Italy). Fluorescein isothiocyanate (FITC) or phycoerythrin (PE)-conjugated monoclonal antibodies specific for porcine CD45 was from LifeSpan Biosciences (Seattle, Washington), for porcine CD11b from BioLegend (San Diego, California), for porcine CD90 from BD PharMingen (San Diego, California), and porcine CD105 and CD29 from Acris Antibodies (Herford, Germany). Appropriate isotype-matched controls from BioLegend (San Diego, California). (3-(4, 5-Dimethylthiazol-2-yl)-2, 5-Diphenyltetrazolium Bromide), MTT was from Sigma Aldrich (Milan, Italy).

Osteogenic and adipogenic differentiation medium: Minimum Essential Medium Eagle (α-MEM) from Lonza (Veuviers, Belgium), Fetal Calf Serum (FCS) from Euroclone (Milano, Italy), dexamethasone, ascorbic acid, β-glycerol phosphate, insulin, indomethacin, 1-methyl-3-isobutylxanthine were all from Sigma-Aldrich (Milan, Italy)). Alizarin Red and Oil Red from Biotica (Milan, Italy). Phosphate-buffered saline (PBS), pH 7.4 was from Euroclone (Milan, Italy). Sterile distilled water was from Braun (Braun, Melsungen, Germany). If not reported, all reagents and solvents used were analytical grade.

### 4.2. Methods

#### 4.2.1. Polymeric Patches Preparation

Polymeric patches were prepared using PLA-PCL copolymers at molar ratio 85:15 and 70:30. The copolymers were selected for their biocompatibility, biodegradability, and mechanical properties. The PLA polymer provides suitable degradation time considering the long timing for cellularization and following in vivo implantation, while PCL component gives plasticity and optimal tensile features. Patches were prepared using two different techniques, temperature-induced precipitation (TIP) and electrospinning (EL), in order to produce prototypes having specific structural features with regard to pore size, pore orientation, porosity, and pore distribution, and to investigate on their impact on their cellularization by p-MSCs.

##### Temperature-Induced Precipitation (TIP) Technique

TIP is a versatile and easy technique for producing highly porous films (TIP-Fs) with mixed porosity, pore size, and pore size distribution and pore orientation.

The bilayer patches were prepared following the protocol reported in a previous work [35]. Briefly, first layer was obtained by dissolving PLA: PLC 85:15 in dioxane (15% *w*/*v*) under magnetic stirring at 500 rpm, in a water/ice bath for 60 min; following the polymer solution was sonicated (3 min) to eliminate the entrapped air bubbles. The solution (2.2 mL) was homogeneously dropped in a Teflon mold (8 cm width × 3 cm length × 4 mm height) and following the system was maintained at −25 °C for 5 h. The prototypes were lyophilized (Freeze dryer Lio-5P, Cinquepascal, Italy) at −48 °C and 0.4 mbar for 12 h.

The second layer was applied on the first lyophilized layer by dropping 2.2 mL of PLA–PCL, 70:30 (10% *w*/*v* in dioxane) solution. Different polymer solution percentages have been used to obtain layers having different porosity and pore size. The lyophilized samples were removed from Teflon mold and they were stored at 4 °C in a controlled humidity environment (RH, 40%).

##### Electrospinning (EL) Technique

Electrospinning (EL) is a well-known fabrication technology based on high electric fields that can be used for producing fibrous scaffolds giving high volume-surface ratio, high porosity, reduced pore size, and structure similar to native ECM [36,37].

Bilayer electrospun matrices (EL-Ms) were obtained using Electrospinning Apparatus NANON-01 (MEEC Instruments, Ogori-shi, Fukuoka, Japan), equipped with dehumidifier (MEEC instruments, MP, Pioltello, Italy). The first layer has been produced electrospinning PLA-PCL 85:15 solution (20% *w*/*v*) in solvent mixture based on DCM: DMF, at ratio 70:30% *v*/*v*. The solvent mixture ratio, polymer concentration, and electrospinning process parameters have been optimized in a previous work as follows: voltage 20 kV, flow rate 0.1 mL/h, relative humidity (RH) 30 ± 5%, temperature 25 ± 4 °C, distance needle-collector 15 cm, collector type metal plate (10 × 10 cm), and electrospinning timing 20 min [20].

PLA: PCL 70:30 polymer solution was electrospun above the first polymeric layer using the same process parameters. The bilayer electrospun matrices (EL-Ms) were located under laminar flow hood to improve solvent evaporation; the dried prototypes were stored at 5.0 ± 2.0 °C in controlled humidity environment (RH, 40%). The bilayer EL-Ms were spherical in shape with a diameter of 80 ± 5 mm.

#### 4.2.2. Prototypes Sterilization

Both types of patches (TIP-Fs and EL-Ms) were exposed to sterilizing ionizing irradiation in order to get sterilized product as required for biomaterials to be implanted in the human body. Sterilized TIP-Fs and EL-Ms have been characterized in the interests of evaluating the impact of γ-sterilization on physico-chemical properties. Mw, Mn, and PI changes were evaluated by gel permeation chromatography (GPC) immediately after the sterilization process (time zero) and data compared with no sterilized samples (controls) in order to evaluate their stability to sterilization process. Guidelines reported in Ph. Eur. 9th Ed, volume III Chapter 5.1 concerning the use of γ-irradiation were followed [38,39]; several papers were published by authors regarding the effect of γ-irradiation on synthetic polymers based on PCL, PLA, PLGA, and related block copolymers [40,41,42,43,44].

Each sample was located in aluminum pounces (polyamide/aluminium/polyester peel for γ-sterilization, VWR) and hermetically closed; the irradiation was carried on at Laboratorio di Energia Nucleare Applicata (LENA) University of Pavia, Italy, using Cobalt-60 as γ-rays source at total irradiation dose of 25 kGy. All the following characterization tests were carried out on sterilized samples.

#### 4.2.3. p-MSC Expansion and Characterization

p-MSCs were obtained from porcine bone marrow aspirate, expanded and characterized as previously reported [45]. Briefly, 20 mL of bone marrow (BM) aspirates were retrieved from 6-month-old piglets under general anesthesia with 3% isoflurane, from the posterior iliac crest. Mononuclear cells (MNCs) were isolated by density gradient centrifugation (Ficoll) and cells were cultured on uncoated polystyrene culture flask at density of 160 kcells/cm^2^ in complete medium at 37 °C, 5% CO_2_. Culture medium was replaced twice a week. Cells were detached by trypsinization at confluence was ≥80%, and then they were counted and re-seeded at a concentration of 4000 cells/cm^2^ for expansion, until passage 4.

At this passage, p-MSCs were phenotypically characterized by flow cytometry using FITC or PE-conjugated monoclonal antibodies specific for porcine CD45, CD11b, CD90, CD105, and CD29. Appropriate isotype-matched controls were added. Analysis of cell populations was performed by direct immunofluorescence with Navios flow cytometer (Beckman Coulter, Milano, Italy); p-MSCs ability to differentiate into osteoblasts and adipocytes was also evaluated. Briefly, to induce osteogenic differentiation, cells were cultured in α-MEM supplemented with 10% FCS, dexamethasone (10^−7^ M), and ascorbic acid (50 μg/mL) β-glycerol phosphate (5 mM). For adipogenesis, insulin (10 mg/mL), indomethacin (0.25 M), and 1-methyl-3-isobutylxanthine (50 mM) were added as well. Media were replaced twice per week for 3 weeks. Osteogenic differentiation was detected by staining calcium depositions with Alizarin Red, while adipogenic differentiation was evaluated by staining fat droplets with Oil Red. The p-MSC phenotypical characterization has been also performed after 30 day of culture on patches, p-MSC were detached from patches by trypsinization, and phenotypical characterized by flow cytometry as previously reported.

#### 4.2.4. Patch Physicochemical Characterization

##### Scanning Electron Microscopy (SEM)

TIP-Fs and EL-Ms morphology was characterized by scanning electron microscopy (SEM). Thickness, pore size, and structure of patches were determined by cross-section SEM analysis. A Zeiss EVO MA10 apparatus (Carl Zeiss, Oberkochen, Germany) at 5 kX, 10 kX, and 30 kX magnifications were used. All samples for SEM were dried and gold sputtered before analysis.

##### Contact Angle (θ) Measurements

TIP-Fs and EL-Ms surface wettability was analyzed in order to evaluate the interaction with aqueous fluids and culture media commonly used in cell culture. Phosphate-buffered saline (PBS), cell culture medium (DMEM LG) and artificial saliva solution (ASS) were used to measure patches contact angle (θ) by DMe-211 contact angle meter apparatus equipped with FAMAS 5.0.2 software (Kyowa Interface Science Co.Ltd, licensed by Exacta Optech, San Prospero, Modena, Italy). The contact angle has been determined by a time-dependent method, and data were expressed as mean ± SD (*n* = 3).

##### In Vitro Degradation Test

In vitro degradation tests were performed in two different experimental sets: (i) on TIP-Fs and EL-Ms prototypes incubated in DMEM-LG p-MSCs medium at 37 °C and (ii) on TIP-Fs and EL-Ms prototypes incubated in ASS at 37 °C after incubation with p-MSCs for 30 days. The degradation tests were aimed to evaluate if and how the different incubation media could affect their degradation behavior and stability.

Regarding the first set of experiments, each sample was placed in 6-multiwell plate (Cellstar^®^, Greiner Bio-one) and soaked in 2 mL Modified Eagle Medium Low Glucose (DMEMLG) 10%, at pH 8.0, 37 °C, 5% CO_2_ and 50% RH for 60 days. Culture medium was changed every 48 h with fresh media; pH shift was monitored along the test using pHmeter 827 pH lab (Metrohm, Switzerland).

In vitro degradation test in ASS was performed on TIP-Fs and EL-Ms prototypes after their incubation with p-MSCs. Briefly, TIP-Fs and EL-Ms were incubated with p-MSCs in DMEM-LG 10% for 30 days at 37 °C, 5% CO_2_, the medium was changed every 48 h. At the end of the incubation time, the cells were detached from each sample using Trypsin solution (1 mL/sample). After cells detachment, they were washed with PBS buffer solution and immersed in ASS (2 mL), which was changed every 48 h with fresh ASS. The samples were further incubated at 37 °C and RH 50% for 60 days. At scheduled timings (30, 45, and 60 days) ASS was collected for pH shifts investigation, while water uptake (WU) and mass loss (ML), weight average molecular weight (Mw), number average molecular weight (Mn), and polydispersity index (PI) changes were evaluated on TIP-Fs and EL-Ms samples.

For Mw, Mn, and PI and ML percentage determination, at scheduled times all samples were collected and washed twice with deionized water to remove ASS residuals. Following, they were frozen at −25 °C for 5 h and lyophilized (Freeze dryer Lio-5P, Cinquepascal, Italy) at −48 °C and 0.4 mbar for 12 h. The lyophilized samples were characterized by further analyses.

##### Gel Permeation Chromatography

GPC analysis was performed with 1260 Infinity GPC (Agilent Technologies, Santa Clara, USA) equipped with pre-column (PLGEL 5 μm), and three columns (PLGEL 5 μm–500 Å; PLGEL 5 μm–103 Å; PhenoGEL 5 μm–104 Å), a pump system (Agilent Technologies 1260 Infinity), and RI detector (Agilent Technologies 1260 Infinity). OpenLab and Cirrus software’s were used for data processing. Samples for GPC analysis prepared by dissolving in THF at concentration of 1–2 mg/mL; each solution was filtered on Whatman Uniflo membrane, 0.45 μm (GE Healthcare, Pittsburgh, USA) before injection. GPC eluent was tetrahydrofuran at a flow rate 1 mL/Min. The weight average molecular weight (Mw) of each sample was calculated using monodisperse polystyrene standards, Mp 580–316,500 Da (Calibration Curve: LogM = 12.26 − 0.3704X^1^ and Coefficient of determination 0.999, Standard Y error Estimate 0.021). The data were processed as weight average molecular weight (Mw), number average molecular weight (Mn), and polydispersity index (PI). Data were expressed as mean ± SD (*n* = 3).

#### 4.2.5. Biological Characterization

All patches used were sterilized by ionizing ΃-radiation at a dose of 25 kGy. p-MSCs (passage 4) were seeded on PLA: PCL 70:30 layer of each prototype and maintained in complete medium at 37 °C 5% CO_2_. Samples without cells were incubated and used as negative controls (Ctr-), while positive control (Ctr+) was represented by the same amount of cells seeded on the bottom of the multiwell plate. Exhausted medium was discarded every two days and it was replaced with fresh medium. Quantitative evaluation of cell viability and proliferation was carried out using MTT assay at scheduled timings (15, 30, 45, and 60 days). Morphological analysis by SEM and DAPI-staining were performed as well. The data obtained from in vitro biological tests were normalized to 1 cm^2^ in order to obtain comparable results independent of patches size.

##### Cell Seeding on Floating Matrices

Square TIP-Fs samples of 5 × 5 mm size and square EL-Ms samples of 10 × 10 mm were used in order to have similar patch weights. Each sample was placed in a 6-multiwell plate and 1 × 10^5^ cells suspended in 50 µL were seeded on surface. After seeding, the samples were incubated at 37 °C, 5% CO_2_ for 10 min in order to promote cells attachment and then 2 mL of complete medium was added into each well. In these experimental conditions, the samples were free to float in the culture medium.

##### Cell Seeding Using CellCrown^TM^ System

TIP-Fs and EL-Ms prototype were cut in a round shape (diameter of 15 mm) in order to be fixed in the CellCrown^TM^ systems (CellCrown^TM^ 12 NX Scaffdex, Tampere, Finland). The CellCrown^TM^ systems were located in 12-multiwell plates and 1 × 10^5^ cells suspended in 50 µL were seeded on the surface. Samples were incubated at 37 °C, 5% CO_2_, and after 10 min, cell incubation medium (2 mL) was added in each well.

##### Cell Viability Determination

Samples were collected at scheduled times and bioavailability was determined by MTT assay. The culture medium was withdrawn and it was replaced with DMEM-LG with no FBS (DMEM-LG *w*/*o* FBS); afterward, 50 µL of MTT solution (5 mg/mL, in DMEM-LG *w*/*o* FBS) was added into each well.

Cells were incubated for 2.5 h at 37 °C to allow MTT reduction by viable cell mitochondrial dehydrogenase. Following this, patches were collected, and a suitable detergent (DMSO) was added to the well for dissolving blue formazan crystals; crystals entrapped into patch were dissolved by the complete solubilization of sample in a proper organic solvent, THF (1 mL).

Multiwell scanning spectrophotometer (Microplate Reader Model 680, Bio-Rad Laboratories, USA) at 570 nm wavelength was exploited for DMSO solutions, while spectrophotometer (Jenway, Steffordshire, UK) at 655 nm wavelength was used for THF solutions. This latter was performed using quartz cuvettes. Results of cell viability were expressed as absorbance of each sample comparing with the absorbance of Ctr+ (cells incubated *w*/*o* patches). The negative control was to verify any interference of biomaterial during analysis. Data were presented as mean ± standard deviation (SD) of six replicates.

##### Cell Fixation Protocol for SEM Analysis

Incubation medium was removed, and the samples were washed with Sodium Cacodilate Buffer (SCB) (0.1 M, pH 7.4) for 7 min (washing step was repeated twice) at each scheduled timing. Cells were fixed with 2% glutaraldehyde (GDA) and 2% paraformaldehyde (PFA) in SCB, fixation incubation was performed for 30 min. Dehydration phase was carried out soaking each sample in ethanol at increasing concentrations (from 30 to 100 *v*/*v*) for 7 min, samples were stored at 4 °C ± 2 between the dehydration phase and the following. Finally, the samples were washed with a mixture (at ratio 50:50) of absolute EtOH 100% *v*/*v* and hexamethyldisilazane (HDMS) for 30 min. SAll samples were dried under laminar flow hood for 12 h and stored in at 4 °C ± 2 until analysis.

##### Staining by DAPI

Staining with DAPI was performed to evaluate adhesion of cells to patches, and to investigate on aggregation and 3D cell organization. Cell nucleus, marked with DAPI, was observed at confocal microscope (Leica, Germany). Samples were removed from incubator at scheduled timings, transferred under chemical hood and three times in PBS heated at 37 °C. PBS was collected and replaced with fixative solution; samples were maintained in contact with fixative solution for 30 min and then they were washed three times with PBS. Following, samples were treated with staining solution (200 μL) and preserved at room temperature for 1–2 h; staining solution was prepared by mixing, in a dark environment, 1 μL of DAPI stock solution with deionized water (199 μL). Samples were washed for further three times with deionized water and they were dried in order to remove excess of water. The timings scheduled for DAPI immunostaining were 15, 30, 45, and 60 days of culture.

#### 4.2.6. Statistical Analysis

Data were expressed as mean ± SD (*n* = 3). Differences in mean values between experimental groups were analyzed by two ways ANOVA (analysis of variance) followed by Tukey’s multiple comparison tests using Graph Pad Prism 7.0 software. A probability value less than 0.05 (* *p* < 0.05) was defined to be significant and 0.01 (** *p* < 0.01), 0.0001 (**** *p* < 0.0001) as highly significant.

## 5. Conclusions

TIP-Fs and EL-Ms exemplify prototypes having different topography and structure; both could be suitable for esophagus regeneration, behaving differently to p-MSC culture. Cell seeding in stating conditions demonstrated to be a suitable method to improve p-MSC cellularization of polymeric patches. Furthermore, the investigation provided interesting information on patches stability in the different experimental conditions that will be further experimented in vivo. Considering the in vitro results, it can be stated that the in vitro protocol proposed for patches cellularization is suitable to achieve homogeneous and complete cellularization of patch. Moreover, the protocol turned out to be simple, repeatable, and reproducible.

The preliminary results collected after in vivo implantation in porcine model are promising; however, further in vivo studies are in progress to statistically prove the data.

## Figures and Tables

**Figure 1 ijms-21-01764-f001:**
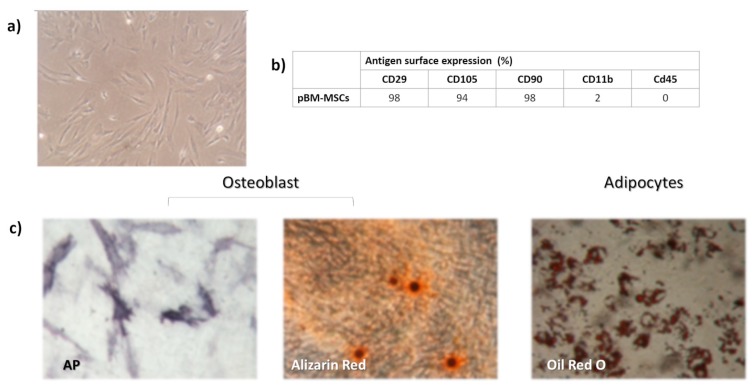
In vitro expanded porcine bone marrow mesenchymal stem cells (pBM-MSC) characterization: (**a**) typical spindle shape morphology of adherent p-MSC; (**b**) percentage of antigen surface expression of pBM-MSCs; and (**c**) osteogenic and adipogenic differentiation capacity after in vitro induction with specific stimuli. Magnification 4× (**a**), 10 and 20× osteoblast and adipocytes, respectively (**c**).

**Figure 2 ijms-21-01764-f002:**
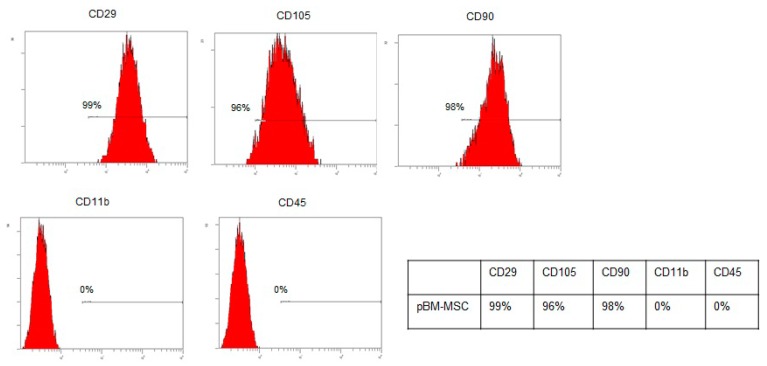
Percentage of antigen surface expression of pBM-MSCs after 30 day of incubation on bilayer electrospun matrices (EL-Ms) patches at 37 °C, 5% CO_2_.

**Figure 3 ijms-21-01764-f003:**
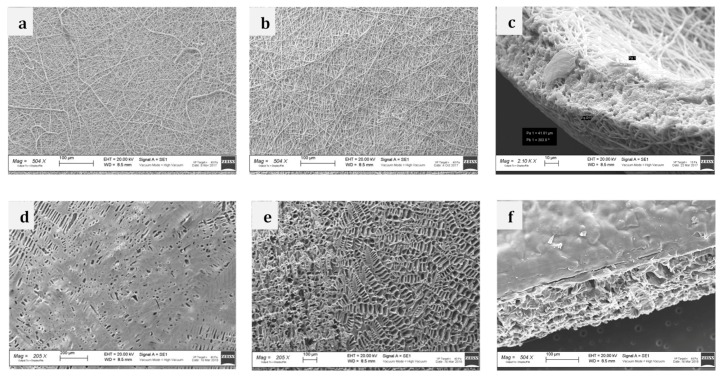
Scanning electron microscopy images of (**a**) EL-M PLA:PCL 85:15 layer; (**b**) EL-M PLA:PCL 70:30 layer; (**c**) EL-M bilayer cross section; (**d**) TIP-F PLA:PCL 85:15 layer; (**e**) TIP-FPLA:PCL 70:30 layer; (**f**) TIP-F bilayer cross section. EL-Ms and TIP-Fs corresponds to bilayer electrospun matrices and temperature induced precipitation porous films, respectively.

**Figure 4 ijms-21-01764-f004:**
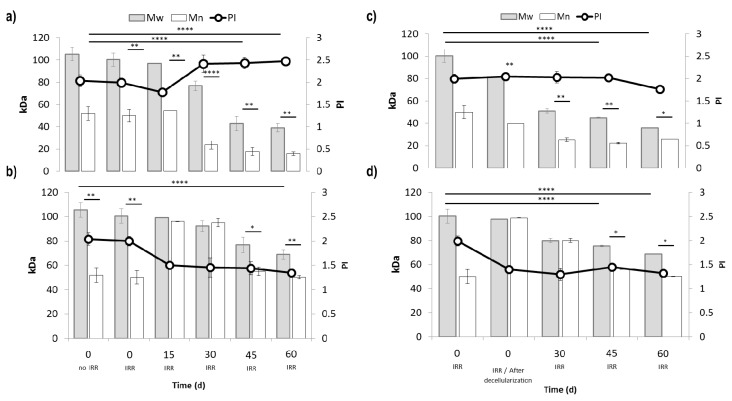
Changes of Mw, Mn, and PI: (**a**) TIP-Fs and (**b**) EL-Ms incubated in Modified Eagle Medium Low Glucose (DMEM-LG) at 37 °C for 15, 30, 46, and 60 days, (**c**) TIP-Fs and (**d**) EL-Ms incubated in Artificial Saliva Solution (ASS) at 37 °C for 30, 45, and 60 days. All samples have been incubated in ASS after incubation with Mesenchymal Stem Cells (MSCs) and decellularization. All Mw, Mn, and PI data have been compared with the corresponding sample non-irradiated and irradiated at time zero. A probability value less than 0.05 (* *p* < 0.05) was defined to be significant and 0.01 (** *p* < 0.01), 0.0001 (**** *p* < 0.0001) as highly significant.

**Figure 5 ijms-21-01764-f005:**
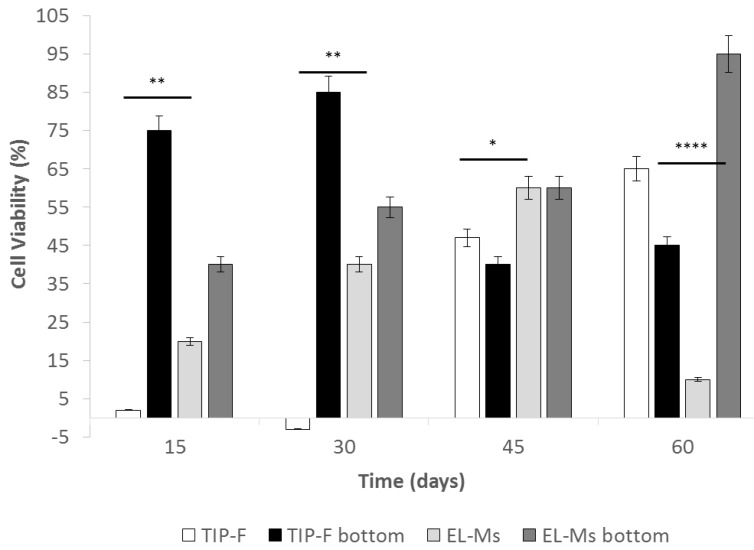
Cell viability percentage of TIP-Fs, EL-Ms incubated with p-MSCs for 60 days, using cells floating protocol. A probability value less than 0.05 (* *p* < 0.05) was defined to be significant and 0.01 (** *p* < 0.01), 0.0001 (**** *p* < 0.0001) as highly significant.

**Figure 6 ijms-21-01764-f006:**
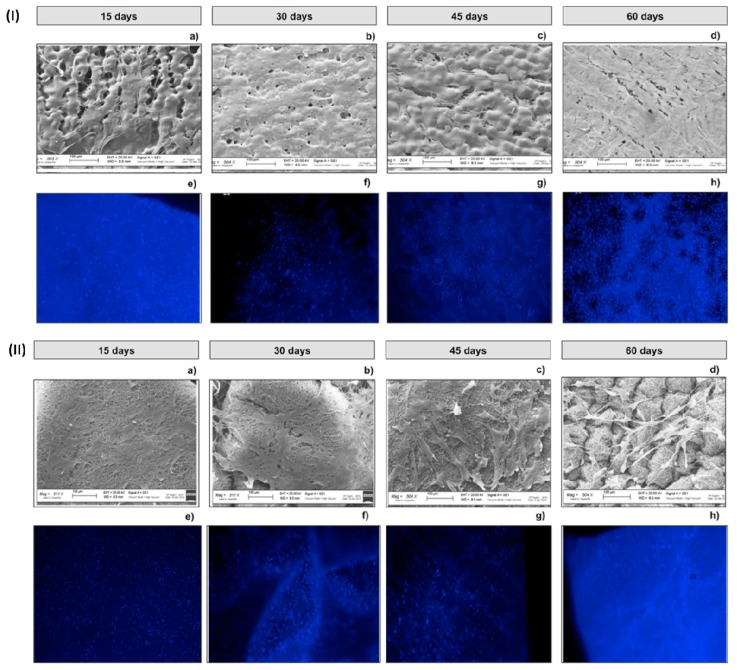
TIP-Fs (**I**) and EL-Ms (**II**) characterization during cellularization in floating conditions. SEM micrographs (**a**–**d**) and confocal images (**e**–**h**). Confocal analyses have been performed after DAPI staining.

**Figure 7 ijms-21-01764-f007:**
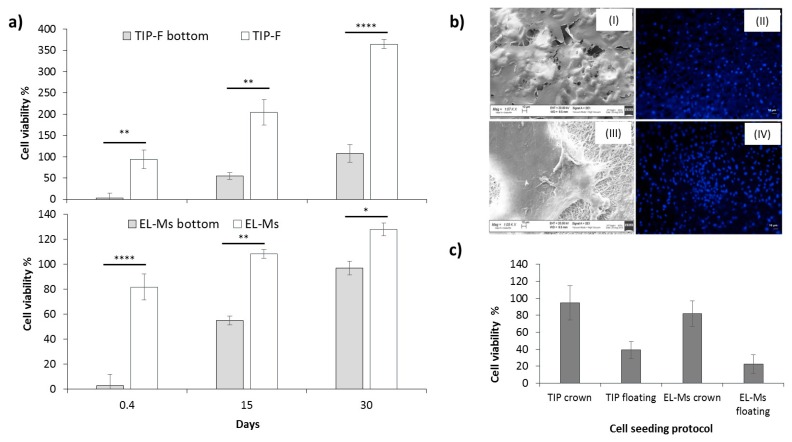
(**a**) Cell viability percentages of TIP-Fs and EL-Ms after incubation with p-MSCs for 0.4, 15, and 30 days and (**b**) cell adhesion of p-MSCs on TIP-Fs and EL-Ms as a function of different cell seeding protocols. (**c**) Cell characterization by SEM and CLSM: SEM (**I**) and confocal (**II**) images of p-MSCs after incubation on TIP-F for 30 days; SEM (**III**) and confocal (**IV**) images of p-MSCs after incubation on EL-Ms for 30 days. Confocal analyses have been carried out after DAPI staining. Magnification SEM analysis 1.07 kX. A probability value less than 0.05 (* *p* < 0.05) was defined to be significant and 0.01 (** *p* < 0.01), 0.0001 (**** *p* < 0.0001) as highly significant.

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
