# Peer review of "Tissue Engineered Esophageal Patch by Mesenchymal Stromal Cells: Optimization of Electrospun Patch Engineering"

_ijms, 2020, doi:10.3390/ijms21051764_

Round 1

Reviewer 1 Report

The authors seek to address a relevant issue in tissue engineering; however, the English Language is pretty poor. The authors must re-write the paper before publication. Also, they should present data on the mechanical property of the patches. Can the authors explain why TIP-Fs cellularization was faster than EL-Ms. Importantly, the author must explain why they decided to evaluate p-MSC static seeding on the patches as free flowing or CellCrown. The real objective of the research is not explicit.

Author Response

Point-by-point reply Reviewer 1

Title

Tissue Engineered Esophageal Patch by Mesenchymal Stromal Cells: Optimization of Electrospun Patch Engineering

Authors

Silvia Pisani, Stefania Croce, Enrica Chiesa, Rossella Dorati*, Elisa Lenta, Ida Genta, Giovanna Bruni, Simone Mauramati, Alberto Benazzo, Lorenzo Cobianchi, Patrizia Morbini, Laura Caliogna, Marco Benazzo, Maria Antonietta Avanzini, Bice Conti.

Abstract

Aim of work was to locate a simple, reproducible protocol for uniform seeding and optimal cellularization of biodegradable patch minimizing the risk of structural damages of patch and its contamination in long-term culture. Two seeding procedures are exploited, namely static seeding procedures on biodegradable and biocompatible patches incubated as free floating (floating conditions) or supported by CellCrownTM insert (fixed conditions) and engineered by porcine bone marrow MSCs (p-MSCs). Scaffold prototypes having specific structural features with regard to pore size, pore orientation, porosity and pore distribution were produced using two different techniques, such as temperature-induced precipitation method and electrospinning technology. The investigation on different prototypes allowed achieving several implementations in terms of cell distribution uniformity, seeding efficiency and cellularization timing. The cell seeding protocol in stating conditions demonstrated to be the most suitable method; these conditions successfully improve the cellularization of polymeric patches. Furthermore, the investigation provided interesting information on patches stability in physiological simulating experimental conditions. Considering the in vitro results, it can be stated that the in vitro protocol proposed for patches cellularization is suitable to achieve homogeneous and complete cellularization of patch. Moreover, the protocol turned out to be simple, repeatable and reproducible.

Reviewer 1

We are grateful to the Reviewer for all comments and suggestions. They help us to improve our manuscript.

  1. The authors seek to address a relevant issue in tissue engineering; however, the English Language is pretty poor. The authors must re-write the paper before publication.

Manuscript has been re-write; english has been improved as suggested by the Reviewer.

  1. Also, they should present data on the mechanical property of the patches.

Line 1667-1668 - Mechanical properties of both patches (TIP-Fs and EL-Ms) have been already published in two separate manuscripts:

Ref 19 - Pisani, S.; Dorati, R.; Conti, B.; Modena, T.; Bruni, G.; Genta, I. Design of copolymer PLA-PCL electrospun matrix for biomedical applications. Reactive and Functional Polymers 2018, 124, 77-89, doi:https://doi.org/10.1016/j.reactfunctpolym.2018.01.011.

Ref 24 - Mechanical properties of TIP-Fs prototype are reported in Ref 24 Dorati, R.; De Trizio, A.; Marconi, S.; Ferrara, A.; Auricchio, F.; Genta, I.; Modena, T.; Benazzo, M.; Benazzo, A.; Volpato, G., et al. Design of a Bioabsorbable Multilayered Patch for Esophagus Tissue Engineering. Macromolecular bioscience 2017, 17, 1600426, doi:10.1002/mabi.201600426.

To ensure that the scaffolds can withstand the physical stresses applied during surgery and the physiological environment, TIP-Fs and EL-Ms prototypes underwent uniaxial mechanical testing. The data discussed in these papers (ref 19 and 24) demonstrated that the fiber structure of EL-Ms prototypes substantially enhances their elastic modulus reaching 21 MPa, while the value resulted for TIP-Fs was around 5 MPa. The high percentage of PCL polymer combined with the fibrous structure boost patch resistance and stiffness; however, the elastic modulus values resulted always to be higher than the physiological maximum pressure esophageal loading, 1.2 KPa. It possible to speculate that the optimal elastic features of EL-Ms prototypes should improve their in vivo performances, in particular during swallowing of food bolus.

  1. Can the authors explain why TIP-Fs cellularization was faster than EL-Ms?

Line 1486 - Cellularization was faster for TIP-Fs than EL-Ms; this evidence could be attributed to structural features that characterize TIP-Fs prototypes such as pore size, pore size distribution and porosity. The proliferation, migration and differentiation of cells within the scaffold are mediated by the initial cell adhesion which depend on pore size, pore size distribution, porosity percentage and pore orientation. The preservation of a balance between the surface area for cell attachment and optimal pore size for cell migration is crucial in the complex cellularization process. 

  1. Importantly, the author must explain why they decided to evaluate p-MSC static seeding on the patches as free flowing or CellCrown.

Line 164 - p-MSC static seeding on patches as free flowing or CellCrown have been investigated in order to define the most suitable experimental conditions for an optimal in vitro cellularization. The seeding procedure and the timing required for the complete cellularization of patches are found out to be crucial affecting the in vivo performances of patch. The final goal was to establish the experimental seeding conditions for the cellularization of TIP-Fs and EL-Ms prototypes before to proceed with their implantation in bovine animal model. Preliminary data collected by the animal study demonstrated that the cellularized patch can be implanted and anastomosed to the native esophageal tissue and the gradually tissue growth over the scaffold occurs. The histological analysis of tissue from pig esophagus post implantation revealed the patch is well and completely integrated with the surrounding tissue with no evidence of inflammatory response. Further studies are in progress and will be object of a further paper focused on porcine animal model.

  1. The real objective of the research is not explicit.

Line 158 - As suggested by the Reviewer the real objective of research has been made explicit in the manuscript.

The project has been developed by a multidisciplinary approach with the aim of optimizing the patch design; their cellularization procedures which involved cells retrieve by biopsy from pig animal, cells expansion and their characterization before seeding and final seeding. The complete patch cellularization has been achieved before implantation in porcine animal model. The seeding phase has been established to be crucial to achieve the complete cellularization of prototype in suitable timing for the following implantation in animal model. Two seeding procedures are exploited in the present research project (p-MSC static seeding on the patches as free flowing or fixed by CellCrown) for the purpose to locate a simple, reproducible protocol for uniform seeding and to minimize the risk of structural damages of patch and its contamination in long-term culture.

Reviewer 2 Report

This paper describes the preparation and performance of tissue patches prepared with a matrix of polylactic(PL)-co-caprolactone(PCL) copolymers into which stem cells are embedded. The performance of the patches is determined by the degradation of the copolymer under pseudo-physiological conditions, which is studied from determination of decrease in the molecular weight by gel permeation chromatography (GPC)

The paper treats a variety of aspects related to biomedical engineering, materials science, etc, which seems of interest for a wide readership. My report concentrates in the physico-chemical characterization of molecular weights.

A general comment : A number of acronyms are employed in this paper. Several of them appear already in the Abstract, without previous definition; this makes the Abstract somehow vain and useless. I suspect that there are some other undefined acronyms across the paper. Perhaps a complete list of all the used acronyms at the beginning of the paper will be helpful.

Lines 84-86 Expressing the composition of copolymers as, e.g., 85:15 “%w/w” - i.e. as if it where ratio PL/PCL of WEIGHTS of the two monomers in the copolymer seemed to me unusual, and I have found it is indeed incorrect. The manufacturer tell that 85:15 is a MOLAR ratio. As the mol. Weight of the monomers are different weight ratio is not the same. They say later (line 11,112) “molar ratio” which is correct.

Linex 31,32 and 84-86 The notation “Mw” seemingly indicates “molecular weight”. Then, line 153 reads “… physicochemical properties Mw, Mn and PI changes…”. I indentify this notation as the typical one to express polymer polydispersity; I realize that Mw is “weight average molecular weigth”, Mn “is number-average molecular weigth” and PI is “polydispersity index (PI=Mw/Mn). The authors forgot to say what Mn and PI are !!

The description of the GPC methodology (Section 3.4.4 lines 223-229) and results is quite incomplete, unclear. I have various doubts and objections:

  • I understand the polymer is extracted from the patches by solubilisation in THF, and the THF solution is injected in the GPC. But what is the eluent used for the chromatograph, THF too or other? Sodium nitrate is a common eluent.
  • The GPC instrument has “… an IR detector”. IR (not defined) seemingly means infrared absorption. I think this is a clear mistake; I have never heard about infrared detection in conventional GPC instruments. The conventional detection is by refractive index (RI), so I suspect this is a mistake (IR should be RI), aggravated by the misuse/abuse of acronyms.
  • GPC analysis with a single detector requires a previous calibration relating elution volume or time to molecular weight. Mention to this, and some details are is missing in the experimental protocol.
  • The results collected from the GPC analysis are Mn and Mw (PI is trivally their ratio). GPC can give more detailed information, particularly the molecular weight distribution of the polymer. I suspect their used Agilent’s OpenLab and Cirrus (line 226) module for SEC/GPC should provide the full distribution curve

There is a Supplementary Material document which just contains a single figure in one page. I don’t see the reason for not integrating thus information within the main body of the paper.

Author Response

Point-by-point reply Reviewer 2

Title

Tissue Engineered Esophageal Patch by Mesenchymal Stromal Cells: Optimization of Electrospun Patch Engineering

Authors

Silvia Pisani , Stefania Croce , Enrica Chiesa , Rossella Dorati * , Elisa Lenta , Ida Genta , Giovanna Bruni , Simone Mauramati , Alberto Benazzo , Lorenzo Cobianchi , Patrizia Morbini , Laura Caliogna , Marco Benazzo, Maria Antonietta Avanzini , Bice Conti

Abstract

Aim of work was to locate a simple, reproducible protocol for uniform seeding and optimal cellularization of biodegradable patch minimizing the risk of structural damages of patch and its contamination in long-term culture. Two seeding procedures are exploited, namely static seeding procedures on biodegradable and biocompatible patches incubated as free floating (floating conditions) or supported by CellCrownTM insert (fixed conditions) and engineered by porcine bone marrow MSCs (p-MSCs). Scaffold prototypes having specific structural features with regard to pore size, pore orientation, porosity and pore distribution were produced using two different techniques, such as temperature-induced precipitation method and electrospinning technology. The investigation on different prototypes allowed achieving several implementations in terms of cell distribution uniformity, seeding efficiency and cellularization timing. The cell seeding protocol in stating conditions demonstrated to be the most suitable method; these conditions successfully improve the cellularization of polymeric patches. Furthermore, the investigation provided interesting information on patches stability in physiological simulating experimental conditions. Considering the in vitro results, it can be stated that the in vitro protocol proposed for patches cellularization is suitable to achieve homogeneous and complete cellularization of patch. Moreover, the protocol turned out to be simple, repeatable and reproducible.

Reviewer 2

This paper describes the preparation and performance of tissue patches prepared with a matrix of polylactic (PL)-co-caprolactone (PCL) copolymers into which stem cells are embedded. The performance of the patches is determined by the degradation of the copolymer under pseudo-physiological conditions, which is studied from determination of decrease in the molecular weight by gel permeation chromatography (GPC).The paper treats a variety of aspects related to biomedical engineering, materials science, etc., which seems of interest for a wide readership. My report concentrates in the physico-chemical characterization of molecular weights.

We are grateful to the Reviewer for all comments and suggestions. They help us to improve our manuscript.

  1. A general comment: A number of acronyms are employed in this paper. Several of them appear already in the Abstract, without previous definition; this makes the Abstract somehow vain and useless. I suspect that there are some other undefined acronyms across the paper. Perhaps a complete list of all the used acronyms at the beginning of the paper will be helpful.

Line 20 - The abstract has been deeply revised, no abbreviations have been included in the abstract moreover.

Abstract - Aim of work was to locate a simple, reproducible protocol for uniform seeding and optimal cellularization of biodegradable patch minimizing the risk of structural damages of patch and its contamination in long-term culture. Two seeding procedures are exploited, namely static seeding procedures on biodegradable and biocompatible patches incubated as free floating (floating conditions) or supported by CellCrownTM insert (fixed conditions) and engineered by porcine bone marrow MSCs (p-MSCs). Scaffold prototypes having specific structural features with regard to pore size, pore orientation, porosity and pore distribution were produced using two different techniques, such as temperature-induced precipitation method and electrospinning technology. The investigation on different prototypes allowed achieving several implementations in terms of cell distribution uniformity, seeding efficiency and cellularization timing. The cell seeding protocol in stating conditions demonstrated to be the most suitable method; these conditions successfully improve the cellularization of polymeric patches. Furthermore, the investigation provided interesting information on patches stability in physiological simulating experimental conditions. Considering the in vitro results, it can be stated that the in vitro protocol proposed for patches cellularization is suitable to achieve homogeneous and complete cellularization of patch. Moreover, the protocol turned out to be simple, repeatable and reproducible.

Line 39 - abbreviation list has been moved at the beginning of the manuscript as suggested by the Reviewer.

  1. Lines 84-86 expressing the composition of copolymers as, e.g., 85:15 “%w/w” - i.e. as if it where ratio PL/PCL of WEIGHTS of the two monomers in the copolymer seemed to me unusual, and I have found it is indeed incorrect. The manufacturer tell that 85:15 is a MOLAR ratio. As the mol. Weight of the monomers are different weight ratio is not the same. They say later (line 11,112) “molar ratio” which is correct.

Line 175 - We agree with the Reviewer, 85:15 is the molar ratio. The typo has been corrected as suggested by the reviewer

  1. Linex 31,32 and 84-86 The notation “Mw” seemingly indicates “molecular weight”. Then, line 153 reads “… physicochemical properties Mw, Mn and PI changes…”. I indentify this notation as the typical one to express polymer polydispersity; I realize that Mw is “weight average molecular weigth”, Mn “is number-average molecular weigth” and PI is “polydispersity index (PI=Mw/Mn). The authors forgot to say what Mn and PI are !!

Line 704 - Mw, Mn and PI have been properly defined in the manuscript. Mw is weight average molecular weight and it takes into account the molecular weight of a chain in determining the contribution to the molecular with average, the number average molecular weight (Mn) is the statistical average molecular weight of all the polymer chains in the sample and PI is used as measure of the broadness of a molecular weight distribution of the polymer. This latter (PI) has been included in the abbreviation list along with Mw and Mn.

  1. The description of the GPC methodology (Section 3.4.4 lines 223-229) and results is quite incomplete, unclear. I have various doubts and objections:

- I understand the polymer is extracted from the patches by solubilisation in THF, and the THF solution is injected in the GPC. But what is the eluent used for the chromatograph, THF too or other? Sodium nitrate is a common eluent.

Patches has been completely dissolved in THF which is the mobile phase (eluent) used for GPC analysis. The selection of THF has been assessed considering columns compatibility and polymer solubility.

- The GPC instrument has “… an IR detector”. IR (not defined) seemingly means infrared absorption. I think this is a clear mistake; I have never heard about infrared detection in conventional GPC instruments. The conventional detection is by refractive index (RI), so I suspect this is a mistake (IR should be RI), aggravated by the misuse/abuse of acronyms.

We agree with the Reviewer, the GPC system is equipped with RI detector. The typo has been corrected as suggested by the reviewer.

- GPC analysis with a single detector requires a previous calibration relating elution volume or time to molecular weight. Mention to this, and some details are is missing in the experimental protocol.

Line 621 - All specification about standards and calibration curve have been included in the manuscript.

GPC analysis was performed with 1260 Infinity GPC (Agilent Technologies, Santa Clara, USA) equipped with pre-column (PLGEL 5 μm), and three columns (PLGEL 5 μm – 500 Å; PLGEL 5 μm – 103 Å; PhenoGEL 5 μm – 104 Å), a pump system (Agilent Technologies 1260 Infinity), and RI detector (Agilent Technologies 1260 Infinity). Cirrus software’s was used for data processing. Samples for GPC analysis were prepared by dissolving in tetrahydrofuran (THF) in order to get final polymer at concentration of 1-2 mg/mL; the each solution were filtered with on Whatman Uniflo membrane (0.45 μm, GE Healthcare, Pittsburgh, USA) before injection. GPC eluent was THF at a flow rate 1mL/Min. The weight average molecular weight (Mw) of each sample was calculated using monodisperse polystyrene standards, Mw 665 – 318,500 Da (Calibration Curve: LogM = 12.26 – 0.3704X1 and Coefficient of determination 0.999, Standard Y error Estimate 0.021). The data were processed as weight average molecular weight (Mw), number average molecular weight (Mn), and polydispersity index (PI). Data were expressed as mean ± SD (n = 3).

- The results collected from the GPC analysis are Mn and Mw (PI is trivally their ratio). GPC can give more detailed information, particularly the molecular weight distribution of the polymer. I suspect their used Agilent’s OpenLab and Cirrus (line 226) module for SEC/GPC should provide the full distribution curve.

We agree with the reviewer, Agilent’s OpenLab and Cirrus provides full distribution curve: Mw (weight average molecular weight), Mn (number average molecular weight), Mz, Mz+1 (higher average molecular weights), and Mp (molecular weight at the highest peak) and PI (polydispersity index).  For all synthetic polydisperse polymers: Mn < Mw < Mz < Mz+1. The polydispersity index is used as a measure of the broadness of a molecular weight distribution of a polymer.

Cirrus software’s was used for data processing and Mw, Mn and PI data have been discussed.

  1. There is a Supplementary Material document which just contains a single figure in one page. I don’t see the reason for not integrating thus information within the main body of the paper.

Line 912 -The Supplementary Material, included in the original version of manuscript, has been included in the manuscript as suggested by the Reviewer.

Round 2

Reviewer 1 Report

The authors made substantial improvement. Accept.